# TRENDy: Temporal Regression of Effective nonlinear Dynamics

**Matthew Ricci, Guy Pelc, Zoe Piran** & **Noa Moriel**
School of Computer Science & Engineering
The Hebrew University of Jerusalem
9190401 Jerusalem, Israel
`matthew.ricci@mail.huji.ac.il`

**Mor Nitzan**
School of Computer Science & Engineering
Racah Institute of Physics
Faculty of Medicine
The Hebrew University of Jerusalem
9190401 Jerusalem, Israel
`mor.nitzan@mail.huji.ac.il`

## Abstract

Spatiotemporal dynamics pervade the natural sciences, from the morphogen dynamics underlying patterning in animal pigmentation to the protein waves controlling cell division. A central challenge lies in understanding how controllable parameters induce qualitative changes in system behavior called bifurcations. This endeavor is particularly difficult in realistic settings where governing partial differential equations (PDEs) are unknown and data is limited and noisy. To address this challenge, we propose TRENDy (Temporal Regression of Effective Nonlinear Dynamics), an equation-free approach to learning low-dimensional, predictive models of spatiotemporal dynamics. TRENDy first maps input data to a low-dimensional space of effective dynamics through a cascade of multiscale filtering operations. Our key insight is the recognition that these effective dynamics can be fit by a neural ordinary differential equation (NODE) having the same parameter space as the input PDE. The preceding filtering operations strongly regularize the phase space of the NODE, making TRENDy significantly more robust to noise compared to existing methods. We train TRENDy to predict the effective dynamics of synthetic and real data representing dynamics from across the physical and life sciences. We then demonstrate how we can automatically locate both Turing and Hopf bifurcations in unseen regions of parameter space. We finally apply our method to the analysis of spatial patterning of the ocellated lizard through development. We found that TRENDy's predicted effective state not only accurately predicts spatial changes over time but also identifies distinct pattern features unique to different anatomical regions, such as the tail, neck, and body–an insight that highlights the potential influence of surface geometry on reaction-diffusion mechanisms and their role in driving spatially varying pattern dynamics.

## 1 Introduction

Nonlinear partial differential equations (PDEs) govern fundamental processes in nature, from the protein waves underlying bacterial cell division (Meinhardt & de Boer, 2001) to the morphogen dynamics giving rise to spatial patterns in animal skin (Kondo et al., 2009). Often, the important aspects of these equations lie not in the fine-grained detail of their solutions but rather in their low-dimensional, qualitative behavior, sometimes referred to as their *effective dynamics* (Wilson, 1965; Kupiainen, 2015; Vlachas et al., 2022); Will the cell divide or not? Will the animal have stripes or spots? And, crucially, how do these effective dynamics change or "bifurcate" as a function of PDE parameters? Understanding the relationship between effective dynamics and system parameters is a major scientific challenge, and analytical approaches can struggle in the

presence of strong nonlinearities (Guckenheimer & Holmes, 1983) and scale dependencies (Binney et al., 1992) commonly found in real systems.

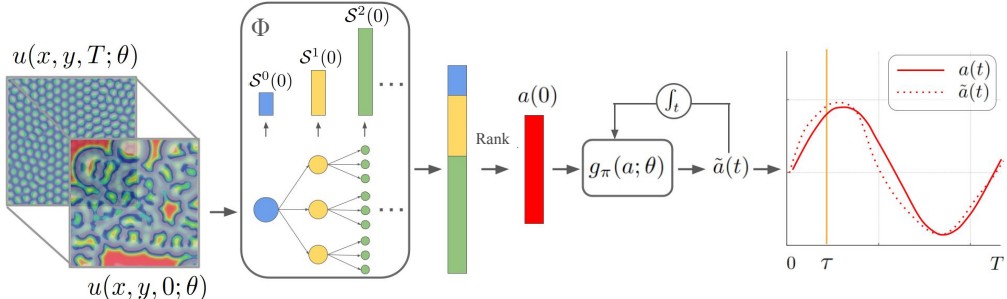

Figure 1: *TRENDy*. *(Left)* Observed dynamics are solutions to PDEs, $u_t = N[u(x, y); \theta]$, with parameters $\theta$ and states $u(x, y)$ taking values on a square domain $\Omega \subset \mathbb{R}^2$. Spatial features are measured at each $t$ using recursive, multiscale filtering via scattering, $\Phi$ (see Secs. 3, A.1.3), and stacked into a single representation, $(\mathcal{S}^0(t) : \mathcal{S}^1(t) : \mathcal{S}^2(t), \ldots)$, where superscripts represent the order of recursion. $\Phi$ thus maps $u$ to reduced-order trajectories which can be further reduced by ranking and subsampling (red). This "effective" representation is controlled by unknown temporal dynamics, $\dot{a}$. *(Middle)* Those unknown dynamics are modeled as a neural network, $g_\pi$, having learnable weights, $\pi$, and which depend on the known, true parameters, $\theta$. *(Right)* Simulated trajectories from the estimated effective dynamics, $\tilde{a}(t)$, are initialized on the true $a(0)$ (using the initial scattering coefficients $\mathcal{S}^i(0)$) and regressed against true effective trajectories, $a(t)$, with a pointwise loss, $L$.

Recently, data-driven approaches have emerged as a promising supplement to analytical methods in the study of PDEs (Rudy et al., 2017), and substantial progress has been made in the fine-grained solution of both forward and inverse PDE problems from data (Yang et al., 2021; Li et al., 2021). However, comparatively little attention has been given to learning effective PDE dynamics and less still to the parametric case. For example, applied Koopman analysis seeks to explain nonlinear dynamics in terms of the behavior of a few coherent spatiotemporal modes, but it does so at the cost of increasing, rather than decreasing, the dimensionality of the state space (Folkestad et al., 2020). Neural operator methods have been previously applied to parametric models (Yang et al., 2021), but focused on fine-grained reconstruction, rather than effective modeling. Work by Vlachas et. al. (Vlachas et al., 2022), for their part, modeled effective dynamics of PDEs with multiscale features and recurrent neural networks but the learned effective dynamics had no relation to system parameters. Lacking a tool to extract the effective dynamics from PDEs in a parametrically interpretable way, the ability to identify and control nonlinear systems is fundamentally limited.

A model of effective dynamics should ideally obey a few desiderata. First, following the standard accuracy-complexity trade-off, the model should be able to represent the dynamics of the underlying PDE in a *compact* manner; namely, the phase space of the effective system should be not only finite- but comparatively low-dimensional, while preserving sufficient information about the observed dynamics. Second, a model of effective dynamics should be *predictive*; it should be able to generalize its knowledge of these low-dimensional dynamics to novel parameter and state settings not observed directly from data. Such a capability is desirable for realistic experimental settings where direct observations of the spatiotemporal dynamics are limited and assessing the system's dynamics over the complete configuration space is infeasible. Finally, the phase space of the effective model should be *interpretable*; the relatively few effective phase variables should represent meaningful spatiotemporal variables whose evolution is predicted by the effective model.

In this spirit, we propose TRENDy (Temporal Regression of Effective Nonlinear Dynamics, Fig. 1), a data-driven approach to modeling effective dynamics in parameterized PDEs. TRENDy works by mapping

the infinite-dimensional state of the observed PDE to low-dimensional space via recursive, multiscale measurements. These low-dimensional measurements are then fit with a neural ODE sharing a parameter space with the observed data. Importantly, TRENDy's latent, effective dynamics allow for direct interpretable predictions about spatiotemporal dynamics, circumventing additional generative components, such as a (learned) encoder or a decoder. We demonstrate this capability on data sets taken from several synthetic and real test cases.

Our principal contributions are

- A formal presentation of TRENDy, in particular its use of the scattering transform (Mallat, 2012) for extracting multiscale measurements.
- Experiments indicating TRENDy can predict qualitative, topological changes in the fitted system's behavior, i.e., bifurcations. We demonstrate this for two models of spatiotemporal dynamics, the Gray Scott model and Brusselator, benchmarking performance in several noise and feature conditions.
- Demonstrations that TRENDy can predict and explain the emergence of complex spatial patterns like those hypothesized to appear through morphogenesis in biological tissue.
- An analysis of spatiotemporal data taken from high-resolution videos of the development of the ocellated lizard. These results indicate a link between body geometry and pattern growth and showcase TRENDy's relevance to real, noisy data in an important biological test case [1].

## 2 RELATED WORK

**Learning parameterized dynamics**  From a data scientific perspective, parameterized models are especially useful as they provide a natural means of generalization to new regions of phase space. Classical methods like Galerkin projection (Holmes et al., 2012) or finite difference schemes (Strikwerda, 2004) have been used to derive reduced-order models or solve for system behavior, although these assume prior knowledge of the governing equations, which is often not available. More recently, data-driven methods like SINDyCP (Sparse Identification of Nonlinear Dynamical Systems with Control and Parameters (Nicolaou et al., 2023)) have gained prominence. SINDyCP extends the otherwise non-parametric SINDy (Brunton et al., 2016) framework by learning control parameters which can be used to steer the system towards target functions. Deep learning approaches include (Tenachi et al., 2023) who showed how deep reinforcement learning could be used to build parametric models of astrophysical phenomena in an unsupervised manner. None of these methods, however, sought to learn effective or reduced-order models, which limits their applicability to settings with strong noise or partial data.

**Effective modeling of PDEs**  The study of effective dynamics seeks reduced descriptions of complex systems that retain essential features. Classical approaches from physics like coarse-graining (Ehrenfest, 1907) and Wilsonian renormalization methods (Wilson, 1965) have long been used in statistical mechanics to model large systems by averaging out microscopic fluctuations. Early data-driven approaches like the equation-free framework (EFF) (Kevrekidis & Samaey, 2009), the heterogeneous multiscale method (HMM) (Weinan & Engquist, 2003) and the Flow Averaging Integrators (FLAVORS) (Tao et al., 2010) approximate the interaction between observed and effective spatial scales with projective integration. Building on this approach (Vlachas et al., 2022) learned latent variable models that capture key dynamical behaviors, enabling efficient long-term predictions in high-dimensional systems like those resulting in turbulence. Contemporary data-driven approaches to learning effective dynamics, howFever, do not learn parametric models, which limits their applicability to the detection of bifurcations, or qualitative (sometimes catastrophic) changes in system's behavior, our main interest.

---

[1]The TRENDy codebase is available at https://github.com/nitzanlab/trendy.

## 3 METHODS

Our goal is to learn predictive models of data generated by spatially extended, autonomous PDEs of the form

$$u_t = N[u(x, y); \theta] \tag{1}$$

where $N$ is a nonlinear operator, $\theta \in \Theta \subseteq \mathbb{R}^k$ is a vector of parameters, and $u(x, y)$ takes values on the square domain $\Omega \in \mathbb{R}^2$ with periodic boundaries (Fig. 1, left). We assume that Eq. 1 undergoes a bifurcation (see Sec. A.1.1) at $\theta = \theta^*$, whereby an equilibrium point changes stability and the qualitative, long-term behavior of the system changes. We further assume that we only have access to a finite data set of solutions $D = \{u(x, y, t; \theta)\}$ indexed by different parameters, $\theta$ and initial conditions, $u(\cdot, \cdot, 0)$. We seek to build a low-dimensional model of this system which can predict behavior as a function of $\theta$ for $\theta$ such that $u \notin D$ (for more details, see Sec. A.1.2).

We observe that, for any Frechét differentiable operator $\Phi : U \to \mathbb{R}^n$, the image $a(t) = \Phi[u](t) \in A$ is a classically differentiable function of $t$. Hence, $\dot{a}$ exists[2] and $a(t)$ is the solution to

$$\dot{a} = g(a; \theta), \tag{2}$$

an ordinary differential equation sharing the same parameter space as Eq. 1 (Fig. 1, right). We refer to Eq. 2 as the *effective dynamics* and $A$ as the *effective phase space*. The true effective dynamics of Eq. 2 is the reduced-order model we would like to learn but which we only know as an image of our data set $D$.

We approximate the effective dynamics with a neural ordinary differential equation (NODE) (Chen et al., 2018),

$$\dot{\tilde{a}} = \tilde{g}_\pi(\tilde{a}, t; \theta) \tag{3}$$

defined on the effective phase space and having learnable weights $\pi$ (Fig. 1, small rounded rectangle). The NODE is instantiated as a multi-layer perceptron whose weights and biases comprise $\pi$. The dependence of the dynamics on $a$ and $\theta$ is effected by augmenting the input to the NODE to be the vector $[a : \theta]$, where : denotes concatenation. In all experiments, the NODE is a multilayer perceptron with four layers, with 64 hidden units in each layer, and with zero-rectification nonlinearities. It is always initialized on the true $a(0)$.

Each training iteration proceeds first by computing effective initial conditions $a_0 = \Phi[u_0]$ sampled from the data set, $D$. Then, Eq. 3 is solved with those initial conditions and their associated parameters, $\theta$. The resulting solutions, $\tilde{a}$, and their derivatives $\dot{\tilde{a}}(t)$, are compared to their ground truth counterparts, $a(t) = \Phi[u](t)$, by integrating across time:

$$L(a, \tilde{a}) = \frac{1}{T - \tau} \int_\tau^T \|a(t) - \tilde{a}(t)\|_2 + \beta \|\dot{a}(t) - \dot{\tilde{a}}(t)\|_2 \, dt. \tag{4}$$

Here, $\tau$ is a burn-in period and $\beta$ is a regularizer on the derivative term. This loss is minimized across $D$ to approximate the argmin weights $\pi$. The approximate effective dynamics, $\tilde{g}_\pi$, are then evaluated on $\theta$ outside of $D$, particularly $\theta$ for which a bifurcation is expected.

The choice of $\Phi$ is an important consideration in the TRENDy construction. In order to keep TRENDy both expressive and interpretable, we set $\Phi$ to be the scattering transform (Mallat, 2012).The scattering transform is a cascade of multiscale band-pass filters with interleaved nonlinearities and averaging. It is provably stable to smooth deformations of the input signal and tends to capture most of the signal energy in its low-order coefficients. Scattering coefficients are computed by iteratively band-pass filtering, taking a modulus, and averaging. For a family of band-pass filters $\psi_{\lambda_i}$ with hyperparameters, $\lambda \in \Lambda$ together with a low-pass filter, $\phi$, examples of zeroth-order, first- and second- coefficients are computed as

$$\mathcal{S}^0 = u \star \phi, \quad \mathcal{S}^1 = (|u \star \psi_{\lambda_i}| \star \phi)_{\lambda_i \in \Lambda}, \quad \mathcal{S}^2 = (||u \star \psi_{\lambda_i}| \star \psi_{\lambda_j}| \star \phi)_{\lambda_i, \lambda_j \in \Lambda^2}. \tag{5}$$

---

[2]We use dot notation for ODEs and subscript notation for PDEs.

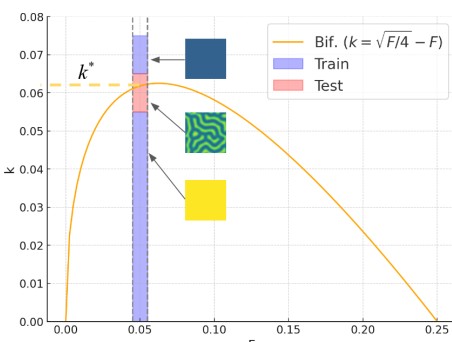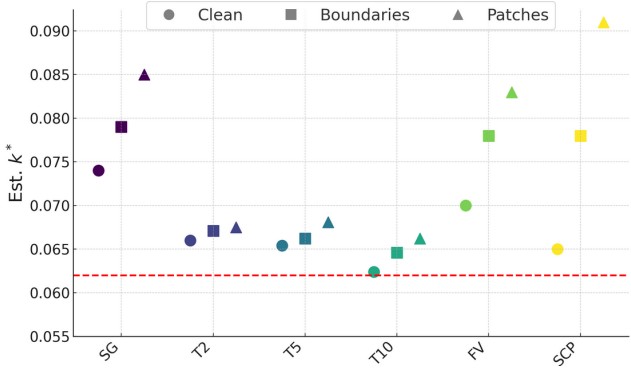

Figure 2: *Bifurcation prediction in the GS model with TRENDy. (Left):* the bifurcation landscape. The GS model transitions to patterning near the orange curve in $F - k$ space. Under the curve, the system is spatially homogeneous ($u \approx 1$, yellow inset). Near the curve, it produces a wide variety of patterns (stripe inset). After the curve, the system is homogeneous again ($u \approx 0$, teal inset). Training data was taken from a thin region of width .01 and centered on $F = .054$. A rectangle of height .01 was held out as test data centered on the bifurcation value of $k^* = .062$. *(Right)*: Detection performance. Numerical continuation was performed on TRENDy trained on five different measurements plus SINDyCP (SG = spatial gradient, T$d$ = TRENDy with $d$ coefficients, FV=Fourier vector, SCP=SINDyCP; details in main text). Estimated $k^*$ values (y axis) are plotted for each of the three noise conditions (Clean as circles, Boundaries as squares and Patches as triangles) for all models. TRENDy with 10 scattering coefficients had the best performance and noise degradation. See main text for details.

We use Morlet wavelets as band-pass filters and a Gaussian for lowpass. Hyperparameters $\lambda_i$ comprise Morlet orientations and scales, and except where otherwise noted we use $\ell = 8$ orientations and $j = 6$ scales. Recursive filtering means that higher-order coefficients are more numerous. Coefficients across orders are stacked into a long vector (Fig. 1, blue, yellow, green). Since many higher-order coefficients tend to be near zero, we take the top $d$ most actived coefficients across time as our representation of input signal (Fig. 1, red bar). The scattering transform is lossy by design but can nevertheless be used for reconstruction given enough coefficients, circumventing the need for a direct decoder when $d$ is large (see Figs. A.5 A.6).

## 4 RESULTS

**Gray Scott** The Gray Scott model (GS) is a simple PDE giving rise to rather complex spatial patterning dynamics (Pearson, 1993). Reaction-diffusion systems, like those of Gray Scott, were investigated in early work by Alan Turing (Turing, 1990) as potential mechanisms for morphogenesis in biological systems, and they remain the subjects of intense theoretical study today (McGough & Riley, 2004; Wang et al., 2016). Nevertheless, it remains challenging to detect reaction-diffusion dynamics in real-world systems, as measurement noise and the preponderance of nuisance variables work to obscure the patterning mechanism.

In that spirit, we sought to showcase TRENDy's ability to predict the emergence of spatial patterns in the Gray Scott model in strong noise conditions. The Gray Scott model describes the evolution of two chemical species via the coupled PDEs:

$$u_t = D_u \nabla^2 u - uv^2 + F(1 - u)$$
$$v_t = D_v \nabla^2 v + uv^2 + (F + k)v \tag{6}$$

which depend on parameters $F, k > 0$ collectively denoted as $\theta \in \Theta$. This system has a spatially homogeneous fixed point at $u = 1, v = 0$ which loses its stability via a pitchfork bifurcation at $k^* = \sqrt{F/4} - F$, producing two new fixed points which are spatially heterogeneous (Fig. 2, left panel, middle inset). Such a bifurcation resulting in patterning is called a Turing bifurcation.

Our goal was to learn a reduced-order model of these dynamics that accurately predicted this Turing bifurcation. We were particularly interested in the robustness of this prediction in three noise conditions: "boundaries", whereby a randomly rotated polygonal boundary having between 4 and 6 sides was used to mask all but the interior of the spatial domain; "patches", whereby between 3 and 6 square patches zeroed out random locations in the dynamics; and "clean", representing the noiseless data (examples in Fig. A.4). Gray Scott parameters (1000 train, 250 test) were drawn from a narrow strip in $F$-$k$ space comprising the rectangle $[0.045, 0.055] \times [0, .075] \subset \Theta$ (Fig. 2, left, shaded regions). Test data was drawn from a small region of length .01 around the bifurcation.

TRENDy was fit to the top 2, 5 or 10 scattering coefficients on average across the data set (conditions T2, T5, T10). We also compared our results to other features, namely, "spatial gradient" (SG): the spatial average of $(\nabla u, \nabla v)$; "Fourier vector" (FV): the Fourier power in 10 wave numbers spaced from the maximum value and the Nyquist limit; and "SINDyCP" (SCP): the aforementioned parametric sparse regression method used for fitting parametric PDEs. The bifurcation for all methods was detected using numerical continuation (further training details in Sec. A.1.5).

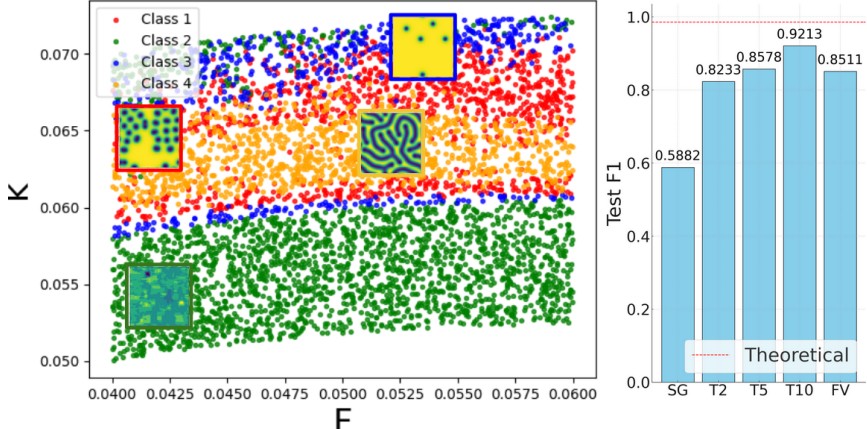

Figure 3: *Predicting effective patterning dynamics*. *(Left)*: true patterning landscape. Samples were generated from the test region shown in Fig. 2 and classified with 4-way k-means using all scattering coefficients (8 orientations, 6 scales, up to order 2). K-means classes (example in insets) were (1) dense spots, (2) homogeneity, (3) sparse spots, and (4) stripes. *(Right)*: A 4-way support vector machine was trained on the predicted state of TRENDy using five types of measurements (see main text). A theoretical maximum F1 score was computed by classifying all features used to generate the original labels (dashed line). F1 scores on test data in each condition are plotted.

After training, we found that TRENDy performed better than competing benchmarks (est. $k^* = .0624$) and degraded gracefully in noisy training conditions (Fig. 2, right). Performance was naturally strongest on the noiseless condition and was comparatively weaker in the boundaries and especially the patches condition. SINDyCP also performed well (est. $k^* = .0651$) in the noiseless condition but degraded sharply with noise. TRENDy performance improved with more coefficients.

In order to demonstrate the interpretability of our approach, we performed a subsequent pattern classification experiment. Understanding the types of patterns that emerge with different parameters is an important area of study in biophysics, systems biology and applied PDEs (Ermentrout, 1991). To that end, we generated a more focused data set within the testing regime of the former experiment and sought to decode pattern classes from the effective dynamics learned by TRENDy in this patterning zone. Ground truth pattern labels were

generated by performing 4-way k-means clustering on the full scattering spectrum of each image (which we confirmed was sufficient to reconstruct pattern quality; see Figs. A.5, A.6). Pattern classes in $F$-$k$ space are shown in Fig. 3.

TRENDy was trained in the same measurement settings (SG, T2, T5, T10 and FV)[3] and its effective steady state was used to decode the true label using a 4-way support vector machine. We found that TRENDy's T10 configuration achieved the highest F1 score on test data. Notably, SG suffices to detect spatial heterogeneity in general (Fig. 2, right, "SG"), but fares poorly on distinguishing different patterning regimes. This confirms that TRENDy's effective coefficients can actually be used to categorically decode back to the observable state space. What's more, we confirmed that scattering coefficients can be directly interpreted to explain spectral distinctions between spots and stripes patterns (see Fig. A.7).

**The Brusselator**    The Gray Scott results were encouraging, but TRENDy was only trained and evaluated in a narrow strip in parameter space. It also sought to model equilibrium dynamics, leaving open the question whether or not our framework need be dynamical at all. To resolve those issues, we used TRENDy to learn an entire bifurcation manifold for the Brusselator model, which exhibits a transition to nonstationary dynamics. The Brusselator is a reaction-diffusion equation used to model chemical oscillations like those observed in the Belousov-Zhabotinsky reaction (Prigogine & Lefever, 1968). Its dynamics are given by the evolution of the concentration of two chemical species:

$$u_t = D_u \nabla^2 u + A + u^2 v - B(1 + u)$$
$$v_t = D_v \nabla^2 v - u^2 v + Bu \tag{7}$$

System behavior is governed by parameters $(A, B) = \theta \in \Theta$. The Brusselator has a stable equilibrium at $\theta = (A, B/A)$ as long as $B < 1 + A^2$, after which a Hopf bifurcation occurs, destabilizing the equilibrium and birthing oscillations. We denote the manifold of Hopf bifurcations by $\gamma = \{(A, B) \in \Theta : B = 1 + A^2\}$. Under periodic boundary conditions, the oscillations in the Brusselator begin as plane waves which can grow into traveling waves which interfere in complex ways.

We sought to test how well TRENDy could learn this basic bifurcation behavior from data, especially under strong noise conditions. To that end, we generated sample solutions to the Brusselator with both the aforementioned noise conditions and a new condition with a holdout factor, $\epsilon$: we designed the training set to only have solutions whose parameters $\theta = (A, B)$ satisfied $d_\gamma(\theta) \equiv \min_{\theta^* \in \gamma} \|\theta - \theta^*\|_2^2 > \epsilon$ for $\epsilon = 0.15, 0.5, 1.0$. The test set always had $d_\gamma(\theta) < \epsilon = 0.15$. We used only the zeroth order coefficients, $S_1$ and $S_2$, which simply measure the average concentration of the species $u$ and $v$ throughout the spatial domain. Since oscillations begin as spatially homogeneous plane waves, we reasoned that a global average would be a sufficient detector. We left coefficients in a linear scale as there was no need to co-register coefficients across several scattering orders. Full training details are in Sec. A.1.5.

We found that TRENDy achieved low forecasting error on test data in the noiseless, low hold-out ($\epsilon = 0.15$) condition which gradually degraded as the noise and hold-out conditions intensified (Table A.2). An example of the trained model exhibiting a Hopf bifurcation is depicted in Fig. 4. Just as in the Gray Scott case, performance was worst in the patches condition and was intermediately worse in the boundaries condition.

Favorable results in solution forecasting do not necessarily imply that our framework accurately predicts the underlying bifurcation structure of the Brusselator. For example, small oscillations around the true steady state would result in a relatively low regression error without accurately matching the intended equilibrium behavior. To investigate this deeper question, we used numerical continuation to locate the estimated bifurcation manifold, $\tilde{\gamma}$, learned by TRENDy in each condition. To that end, for each condition, we ran 100 numerical continuation experiments corresponding to evenly spaced $A \in [1.5, 3.5]$. All points, $\theta$, (if any) determined to mark a Hopf bifurcation as $A$ was varied were collected and a quadratic fit was applied to

---

[3]SINDyCP does not have a latent state, so it does not lend itself naturally to this classification problem.

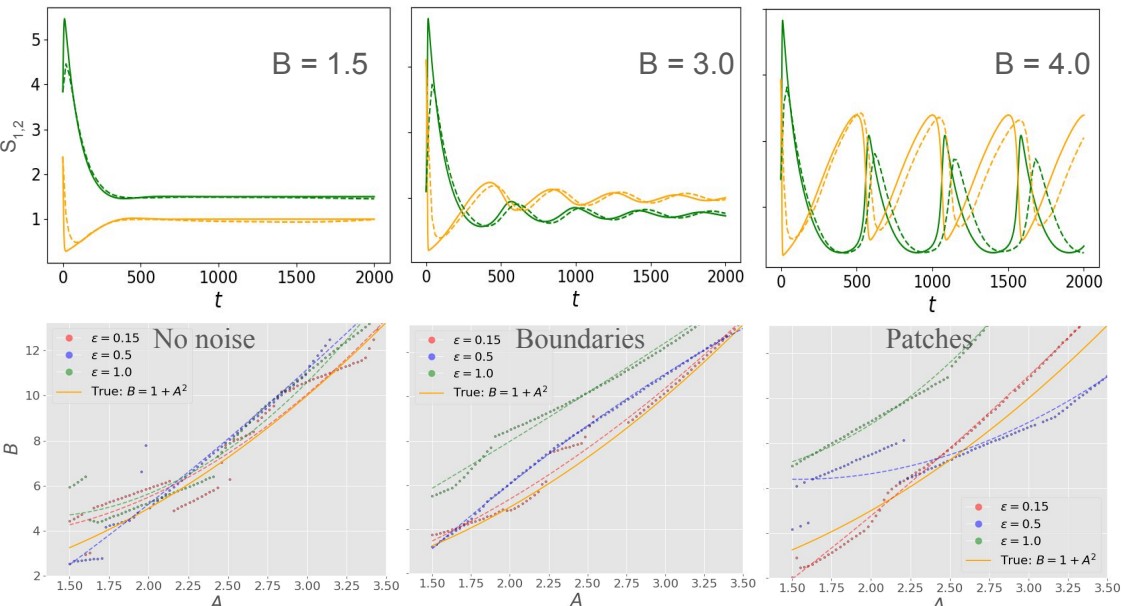

Figure 4: *TRENDy learns reduced-order models of the Brusselator. (Top row)* TRENDy was trained on zeroth-order scattering coefficients, ($S_1$ in solid green, $S_2$ in solid orange) measured from the evolving Brusselator model (Eq. 7). The trained model (here depicted with hold-out parameter $\epsilon = .15$ and without noise) closely predicted the true effective dynamics across the bifurcation boundary (depicted: $A = 1.5$, $B = 1.5, 3.0, 4.0$). *(Bottom row)*. For each $A$ between 1.5 and 3.5 in 100 steps, numerical continuation was performed on the trained TRENDy model. Whenever a Hopf bifurcation was detected at $B$, that point $(A, B)$ was tabulated and plotted. Then, a quadratic polynomial was fit to all points within one holdout ($\epsilon$) and noise condition. The true Hopf manifold is in orange at $B = 1 + A^2$. Approximation of the manifold is qualitatively correct across conditions, but worsens in the boundaries and patches conditions.

estimate $\tilde{\gamma}$. Note that, since the minimum $\epsilon$ used for holdout was $0.15$, TRENDy never observed an actual bifurcation during training.

Mirroring the forecasting results, TRENDy located the true bifurcation manifold most accurately in low-noise, low-holdout conditions and this capability slowly degraded in the more challenging conditions (Fig. 4). The lower panels of Fig. 4 show, for each $\epsilon$, the estimated bifurcation curves $\tilde{\gamma}$ along with the parameters to which they were fit. In the no noise condition, despite some spurious detections, TRENDy provides a good fit to the true manifold. The fit weakens in the noise conditions which also tend to exacerbate the effect of $\epsilon$. The estimated manifold is almost always concave up and is always monotonically increasing, matching the qualitative behavior of the true manifold.

**Spatial patterning in the ocellated lizard**    As a final proof of concept of our method, we trained TRENDy to predict the spatial patterning dynamics of the ocellated lizard (*T. lepidus*). This species of lizard has distinctive skin patterns made from a mosaic of black and green scales (Fig, 5A, upper inset taken from Fofonjka & Milinkovitch (2021)). The dynamics of these patterns through development has been the subject of numerous computational and biological studies (Milinkovitch et al., 2023). Importantly, recently work by Fofonjka & Milinkovitch (2021); Manukyan et al. (2017) used a combination of computational modeling and real data collection using optical high-resolution episcopic microscopy, to build extremely detailed reaction-diffusion models of scale patterning through development. The authors argue that growth and other

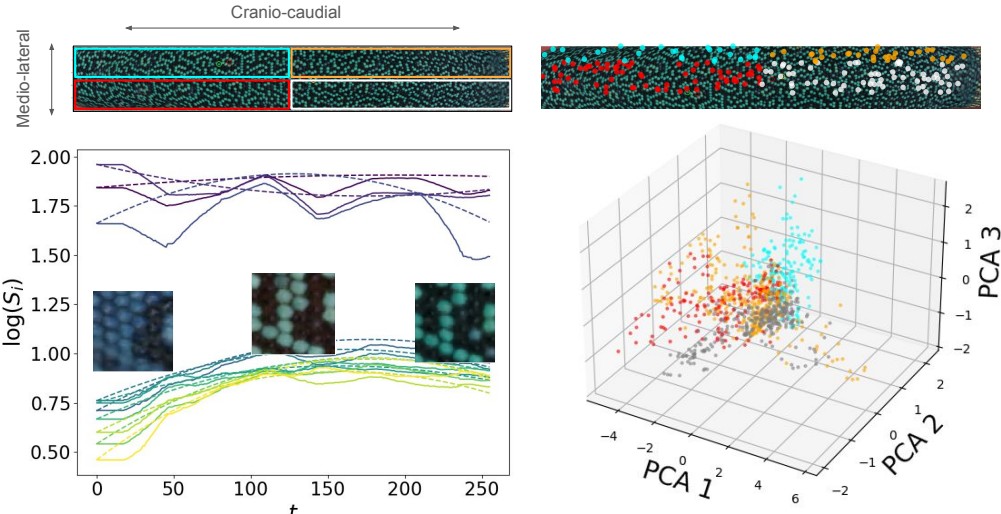

Figure 5: *TRENDy learns scale dynamics of the ocellated lizard. (Left, top inset).* A cropped version of the adult ocellated lizard's torso in medio-lateral, cranio-caudal coordinates. Four regions are highlighted in colors corresponding to their quadrant label in this coordinate system. *(Left, lower).* Dynamics of one patch of scales taken from the origin from juvenile to adult state. Features represented spatially averaged scattering coefficients. Solid lines are the true dynamics and dashed are TRENDy estimates. The insets depict the patch at $t = 0, 125, 250$. *(Right, top inset).* An SVM was trained on the final, 10-d state of TRENDy with labels given by the patch's quadrant. This inset depicts prediction on the test data, where TRENDy achieved 95% accuracy. *(Right, lower).* Together with the classifier results, the moderate clustering of TRENDy states in PCA space indicates a relation between scale dynamics and anatomical coordinates.

mechanical processes influence patterning, much like how work by Murray (1981) explained why stripe emergence in animal fur occurs preferentially in tubular structures like limbs and tails.

In that spirit, we used TRENDy to learn a reduced-order model of the evolution of skin patches of a single ocellated lizard through development from its juvenile to adult stages, and then investigate the trained model for a relation between pattern dynamics and body coordinates.

We first analyzed data generated by Fofonjka & Milinkovitch (2021); Manukyan et al. (2017), and acquired a dataset of patch evolutions by randomly sampling locations in a high-resolution video of the developing ocellated lizard. We fit TRENDy to $d = 10$ scattering coefficients and used patch coordinates (upper left corner) as parameters, $\theta$, of the NODE. More details are found in Sec. A.1.5.

As before, the trained model closely tracked the true coefficients (Fig. 5A, dashed vs solid lines, respectively). In order to observe how TRENDy encoded the underlying animal geometry, we then labeled each sub-video by the quadrant from which it was sampled: in image coordinates, north-west, north-east, south-east or south-west. We then investigated whether quadrant labels could be decoded from the final states, $\tilde{a}(T)$, of the estimated dynamics. If this were to be the case, it would demonstrate that TRENDy had learned a relationship between scale dynamics and the underlying animal geometry. To that end, we did an 80-20 split of the $a(T)$ into training and testing sets and fit an SVM to the training samples.

We found that this SVM had high testing performance (accuracy .95, weighted F1: .95), indicating that TRENDy had learned to differentiate scale dynamics based on patch location and geometry. This is especially

clear if we color patch coordinates according to their labels (Fig. 5B, upper inset) and observe how the colors closely respect quadrant boundaries. PCA on the final states, $\tilde{a}(T)$, also reveals significant clustering by quadrant label (Fig. 5B, lower panel, colors corresponding to quadrant labels in panel A). These results demonstrate that, not only can TRENDy fit dynamical data, but it can also highlight the potential influence of surface geometry on reaction-diffusion mechanisms and their role in driving spatially varying pattern dynamics. We note that scale dynamics have also been modeled with tools from statistical mechanics (Zakany et al., 2022) and we include an application of TRENDy to this setting in Sec. A.1.5, *Ising Model*.

## 5 DISCUSSION

TRENDy, our approach to learning predictive models of spatially extended PDEs, synthesizes techniques from reduced-order modeling ((Vlachas et al., 2022)) with methods from parameterized modeling fitting (Nicolaou et al., 2023). We have demonstrated this approach in benchmarked experiments on both synthetic and real data, and have used it to predict bifurcations and visualize the relation between dynamics and geometry in spatial patterning. Among its advantages, TRENDy's use of hardwired scattering features makes it robust to noise and avoids feature learning through self-supervision, which can be costly. Importantly, we have also shown how existing methods for fine-grained spatial prediction are not necessarily performant on bifurcation and classification problems, especially in the presence of noise.

We envision several areas for improvement upon the current framework's limitations. Notably, though we have demonstrated that scattering coefficients provide an expressive, interpretable and efficient form of effective measurements, they are by no means the only possible choice. Future work could investigate learning these measurements, possibly with the inclusion of an explicit decoder and reconstruction loss. Here, challenges will be twofold: first, the reconstruction of observable dynamics is underdetermined since TRENDy's compression is lossy, though one could regularize decoding with symbolic regression as in Nicolaou et al. (2023); second, learned features may be less interpretable than scattering coefficients. Furthermore, while the current iteration of TRENDy performs well on real data obeying equilibrium dynamics, its application to real oscillatory data is limited. Indeed, high-frequency, noisy data can simply lead to repeated, conflicting learning signals about the equation of motion at a particular state (Oh et al., 2024). Additionally, subsequent studies could investigate frameworks which explicitly account for noise, for example (Frishman & Ronceray, 2020).

While our investigation of spatial patterning in the ocellated lizard is largely a proof of concept, one of our principal interests is the application to real data representing bifurcations in realistic noise settings and sample sizes. For example, various studies in synthetic biology have produced high-resolution videos of spatial dynamics of the Min protein system (Glock et al., 2019; Rajasekaran et al., 2024), an important regulatory mechanism for cell division and morphogenesis. Investigating how this and related systems undergo qualitative changes in behavior is a necessary step for understanding fundamental processes in biology, and will require similar data-driven frameworks for robustly modeling the relation between system parameters and function which we have begun to study here.

## ACKNOWLEDGMENTS

We thank the Nitzan lab for their thoughtful feedback. This work was funded by the European Union (ERC, DecodeSC, 101040660) (M.N.). Views and opinions expressed are, however, those of the author(s) only and do not necessarily reflect those of the European Union or the European Research Council. Neither the European Union nor the granting authority can be held responsible for them.

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

# A   APPENDIX

## A.1.1   PDEs, TOPOLOGICAL CONJUGACY AND BIFURCATIONS

We will focus on PDEs of the form

$$u_t = D\Delta u + f_\eta(u) = N_\theta[u], \tag{A.1}$$

for $f$ a smooth (polynomial) operator on states, $u(r, t)$, taking values on a square domain $D \in \mathbb{R}^2$ with periodic boundaries. The subscript $\theta$ denotes a set of parameters $(D, \eta) \in \Theta \subseteq \mathbb{R}^k$. Assume that the solution to Eq. A.1 with initial condition $u_0 = u(0)$ admits a solution $u$ in the Sobolev space $H^2(D)$ for each $t$.

We are interested in the long-term behavior of solutions, $u$, to Eq. A.1. Let $\phi(t, u_0)$ be the flow associated to the system, which maps an initial condition $u_0$ to its state at time $t$. Then, the $\omega$-limit set of $u_0$ is given by

$$\omega(u_0) = \left\{ u \in H^2(D) \mid \exists \{t_n\} \to \infty \text{ such that } \phi(t_n, u_0) \to u \right\}. \tag{A.2}$$

In other words, $\omega(u_0)$ is the set of all states visited by the system starting at $u_0$ after an infinite amount of time. The set of all $\omega$-limit sets of Eq. 1 for a given parameter $\theta$ is denoted $\Omega_\theta$. It may comprise fixed points, limit cycles, strange attractors or yet more esoteric structures invariant under $\phi$. It is precisely the nature of $\Omega_\theta$ as a function of $\theta$ which we wish to understand and predict from data.

In many cases, changing $\theta$ will have no effect on $\Omega_\theta$. For example, consider two partial differential equations

$$u_t = N_{\theta_1}[u] \tag{A.3}$$
$$u_t = N_{\theta_2}[u],$$

representing two different parameter settings, $\theta_1$ and $\theta_2$. The PDEs $u_t = N_{\theta_1}[u]$ and $u_t = N_{\theta_2}[u]$ are said to be topologically conjugate if there exists a homeomorphism $h : H^2 \to H^2$ such that the following diagram commutes for all $u$ and $t$:

$$h \circ \phi_{\theta_1}(t, u) = \phi_{\theta_2}(t, h(u)),$$

where $\phi_{\theta_1}(t, u)$ and $\phi_{\theta_2}(t, u)$ are the flows for their respective PDEs. In this case, we write $\phi_{\theta_1} \sim \phi_{\theta_2}$. If two systems are topologically conjugate, then their orbits (as functions of $t$) can be mapped homeomorphically to one another so that their dynamics are topologically identical. Importantly, it can be shown that, if $\phi_{\theta_1} \sim \phi_{\theta_2}$, then $\Omega_{\theta_1} = \Omega_{\theta_2}$.

However, if there exists a $\theta^*$ such that, for all $\epsilon > 0$ there exists $\theta$ such that $\|\theta - \theta^*\|_2 < \epsilon$ and $\Omega_\theta \neq \Omega_{\theta^*}$, then we say that the system exhibits a *bifurcation* at $\theta^*$. Informally, the parameter $\theta^*$ is a transition point between two qualitatively different (topologically non-conjugate) behavioral regimes of the system. These transitions may represent the emergence of periodic behavior, spatial patterning or other dynamical regimes. Predicting the location of these bifurcations from data is a focus of this paper.

## A.1.2   REDUCED-ORDER MODELING AND EFFECTIVE DYNAMICS

Reduced order modeling (ROM) aims to reduce the complexity of high-dimensional systems while preserving their effective dynamics. Analytical approaches, such as the method of weighted residuals and Galerkin projection, approximate the solution space by projecting the governing equations onto a lower-dimensional subspace defined by a set of basis functions (Holmes et al., 2012). Data-driven approaches, including proper orthogonal decomposition (POD) and dynamic mode decomposition (DMD), leverage observed data to construct reduced models that capture the dominant features of the system (Brunton et al., 2020). These methods are increasingly integrated with machine learning techniques to enhance model accuracy and generalization (Hesthaven & Ubbiali, 2018) (Vlachas et al., 2022).

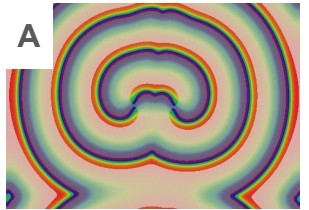 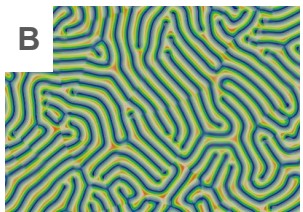 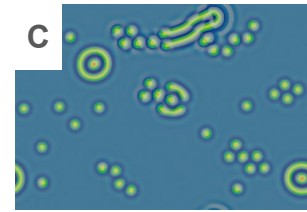

Figure A.1: *Spatial structures in PDEs.* All panels represent solutions to the Gray Scott reaction-diffusion model. *(A)* A traveling wave of period equal to about 1/6 of the width of the spatial domain $D$. Fourier coefficients are a good detector for the Hopf bifurcation leading to these spatial structures. *(B)* However, if we want to localize a Turing bifurcation in terms of both its critical parameter, $\theta^*$ and the location of its new equilibrium, we must be able to resolve spatial structures with features more sophisticated than Fourier modes. For example, stripes like those depicted here can be aliased in Fourier space with spots or other visually different structures. *(C)* Fourier features also struggle to model localized structures like pulses and solitons which occur in systems with strong nonlinearities and coupling across spatial scales. Figures publically available at Munafo.

Our goal in this paper is to investigate which reduced order models lead *generally* to good predictive models of bifurcation structure in PDEs. We will conceive of such models as operators (or *measurements*), $\Phi$, mapping from the observable state space of the original dynamics to a finite- and indeed low-dimensional space of features. That is, we wish to know, given an arbitrary PDE which bifurcates at $\theta^*$, which operators on PDE data lead reliably to an effective model which also bifurcates at $\theta^*$? This bifurcation is the "effect" we want the model to capture.

A few examples are instructive. For instance, imagine that $u_t = N[u; \theta]$ undergoes a Hopf bifurcation at $\theta^*$ whereby an initially stable spatially uniform fixed point, $u_0(r) \equiv c$, loses stability preceding the emergence of uniform (plane) waves. The trivial averaging operator

$$\Phi[u](t) = \frac{1}{|D|} \int_D u(r, t) \, dr \tag{A.4}$$

gives a one-dimensional, spatially global signal $a(t) = \Phi[u](t)$ which is differentiable in $t$ since $\Phi$ is differentiable in the sense of Fréchet. Hence, $a$ is the solution to the ODE

$$a_t = g(a; \theta) \tag{A.5}$$

for some $g$ giving the law of motion of the spatial average as a function of $\theta$. Clearly, Eq. A.5 has a fixed point at $a_0 = \Phi[u_0]$ and undergoes a Hopf bifurcation at the same $\theta^*$ as the underlying PDE. Note that $g$ very rarely has a closed form, but, if we had a good model of $g$, we could reliably predict the onset of plane waves.

Now, imagine instead that the bifurcation at $\theta^*$ leads to the onset of traveling waves with equilibrium position $c$ and having characteristic spatial wave number $k$ (see Fig. A.1 A). As the wave is centered symmetrically about $c$, we have $\Phi[u] \equiv c$ both before and after the Hopf bifurcation; i.e., a simple average does not detect the relevant spatial structure associated to the bifurcation. In this case, a more appropriate measurement would be power associated to a given spatial frequency, $k = \sqrt{k_x^2 + k_y^2}$. For instance, if

$$\hat{u}(k_x, k_y, t) = \mathcal{F}(u)(k_x, k_y, t) \tag{A.6}$$

and we define the spectral density at $(k_x, k_y)$ via

$$P(k_x, k_y, t) = |\hat{u}(k_x, k_y, t)|^2,\tag{A.7}$$

then

$$\Phi_k[u](t) = \int_0^{2\pi} P(k\cos\psi, k\sin\psi, t)k\,d\psi\tag{A.8}$$

is a reasonable detector for periodic spatial structures off frequency $k$. Indeed it is precisely these Fourier coefficients which have been used historically to solve systems like Eq. 1, so we should expect $a_t = \frac{d\Phi_k[u]}{dt} = g(a, \theta)$ to be a good model of the effective dynamics. Even if we don't know the true $k$, using a composite measurement involving a few different Fourier modes would help detect the relevant wave scale while remaining relatively low-dimensional.

Yet, being a spatially global signal, spectral density cannot reliably detect changes in local structure, as illustrated by the emergence of pulses, solitons or other non-periodic structures in bifurcating systems with strong nonlinearities or coupling across spatial scales [cite] (Fig. A.1C). From a pure signal processing perspective, we observe, for example, that the cosine grating

$$I_1(x, y) = \cos(k_x x + k_y y)\tag{A.9}$$

and the windowed grating

$$I_2(x, y) = A\exp\left(-\frac{x^2 + y^2}{2\sigma^2}\right)I_1(x, y)\tag{A.10}$$

have the same spectral density at $k = \sqrt{k_x^2 + k_y^2}$ when $A = \sqrt{\frac{2}{\pi\sigma^2}}$. Hence, any bifurcation heralded by the emergence of this sort of local structure is liable to be missed by a (global) signal like spectral density.

Evidently, we need to rely on a measurement operator which captures information across multiple spatial scales, preferably in an interpretable manner, so that particular spatial structures can be identified as being associated with a bifurcation. One such measurement, investigated in subsequent experiments, is described next.

### A.1.3 THE SCATTERING TRANSFORM

The scattering transform of Mallat and colleagues is a mathematical framework designed to extract stable and informative features from signals or images while preserving critical structures, such as texture and edges, that are typically lost under traditional signal processing methods like the Fourier or wavelet transforms. The scattering transform extends the concept of wavelet transform by incorporating nonlinear operations, enabling it to capture higher-order interactions within a signal.

Formally, the scattering transform is defined as a cascade of wavelet transforms followed by a modulus nonlinearity and an averaging operation. Given a two-dimensional input signal $u(x, y)$, a set of wavelet filters $\{\psi_\lambda\}_{\lambda \in \Lambda}$ is applied, where $\Lambda$ denotes the set of scales and orientations of the wavelets. The wavelet transform of $u$ is then given by the convolution,

$$[u \star \psi_\lambda](x, y) = \int_{\mathbb{R}^2} u(v, w)\psi_\lambda(x - v, y - w)\,dvdw.\tag{A.11}$$

The modulus operator $|\cdot|$ is applied to the transformed signal to introduce nonlinearity, which is crucial for capturing higher-order features:

$$U^{(1)}(\lambda) = |u \star \psi_\lambda|.$$

This operation is iterated to produce higher-order coefficients:

$$U^{(m)}(\lambda_1, \ldots, \lambda_m) = \left| U^{(m-1)}(t, \lambda_1, \ldots, \lambda_{m-1}) \star \psi_{\lambda_m} \right|.$$

The final scattering coefficients are obtained by applying a low-pass filter, typically an averaging operation, to each $U^{(m)}$, yielding the scattering transform:

$$S^{(m)}u = U^{(m)} \star \phi$$

where $\phi$ is a low-pass filter that ensures the invariance of the scattering coefficients to translations of the input signal.

The scattering transform can be discretely implemented as a convolutional network where each layer performs a wavelet transform followed by a modulus operation and subsampling. Given a discretely sampled signal $u[x, y]$, the wavelet filters $\psi_\lambda$ are applied, with $\lambda$ indexing both orientation, $\theta$, and scale, $j < J$. For the case of a Gabor filter, this could be

$$\psi_{j,\theta}[x, y] = \frac{1}{\sqrt{2^j}} \exp\left(i(x \cos\theta + y \sin\theta)\right) \exp\left(-\frac{x^2 + y^2}{2 \cdot 2^j}\right). \tag{A.12}$$

In the discrete implementation, after applying the modulus operator $|\cdot|$, the result is subsampled by a factor $2^j$, where $j \leq J$. This process captures multi-scale, multi-orientation features of the signal, constructing higher-order coefficients through iterative application. The final scattering coefficients are obtained by applying a low-pass filter, typically a convolution with a scaling function ensuring translation invariance.

### A.1.4 NUMERICAL CONTINUATION

Numerical continuation methods aim to trace the set of solutions $x(\mu)$ of a parameterized system of ordinary differential equations (ODEs):

$$\frac{dx}{dt} = f(x, \mu), \quad x \in \mathbb{R}^n, \quad \mu \in \mathbb{R},$$

as the parameter $\mu$ varies. The goal is to compute the evolution of solutions $x(\mu)$ and detect bifurcations (see Sec. A.1.1). A solution $x_0$ at a parameter value $\mu_0$ satisfies the steady-state condition

$$f(x_0, \mu_0) = 0.$$

To compute the continuation of this solution as $\mu$ changes, one solves the augmented system:

$$F(x, \mu) = f(x, \mu) = 0.$$

Starting from a known solution $(x_0, \mu_0)$, numerical continuation extends the solution curve by incrementally varying $\mu$. A typical method for this is the predictor-corrector approach. In the predictor step, a linear approximation to the solution branch is made by using the tangent direction, which can be computed by differentiating $F(x, \mu) = 0$ with respect to $\mu$. In the corrector step, Newton's method (or another iterative solver) is used to refine the approximation so that $f(x, \mu) = 0$ holds exactly.

A bifurcation occurs when the qualitative structure of the solution set changes. This can often be detected by examining the Jacobian matrix $D_x f(x, \mu)$. At a bifurcation point, an eigenvalue of $D_x f(x, \mu)$ crosses

the imaginary axis, indicating a change in stability or the emergence of new solution branches. Numerical continuation algorithms can detect such points by tracking the eigenvalues of the Jacobian as $\mu$ is varied.

When a bifurcation condition, such as the zero crossing of an eigenvalue, occurs within an interval of parameter values, a bisection algorithm can be used to refine the location of the bifurcation point. The bisection method works by iteratively halving the interval and evaluating the bifurcation condition at the midpoint, continuing this process until the bifurcation point is found to within a specified tolerance.

### A.1.5 EXPERIMENTAL DETAILS

Simulated reaction-diffusion data was created by solving a forward Euler scheme on a square domain of side length 64 and spatial discretization $dx = 1.0$. Ising data was solved on the same domain but with a Glauber dynamics Glauber (1963). All scattering coefficients were computed with $L = 8$ orientations and $J = 6$ scales; i.e. up to the maximum scale of $2^6 = 64$.

Scattering coefficients were averaged across the data set and time and subsequently ranked to produce the top $d$ for use in fitting. For benchmarking, we chose $d = 2, 5, 10$. We also compared these representations to

- Spatial gradient (SG): For a solution with channels $u, v$, this is given by

$$\left( \frac{1}{|\Omega|} \int \nabla u(x, y) \ dx \ dy, \frac{1}{|\Omega|} \int \nabla v(x, y) \ dx \ dy \right), \tag{A.13}$$

  where $|\Omega|$ is the area of the square domain. This is meant to detect spatial homogeneity in a very general sense.
- Fourier vector (FV): this is a vector of 10 wavenumbers. To compute the power spectrum at 10 evenly spaced wavenumbers between the Nyquist limit and the maximum spatial scale in a two-dimensional square domain, excluding the DC component, we define $k_{\text{Nyquist}} = \frac{N}{2}$ for $N = 64$, the number of grid points in one dimension of the square domain, and $k_{\min} = \frac{1}{dx \times 64} = \frac{1}{64}$. Then, ten evenly spaced wavenumbers are $k_n = k_{\min} + n \cdot \Delta k, \quad n = 1, 2, \ldots, 10$, where $\Delta k = \frac{k_{\text{Nyquist}} - k_{\min}}{9}$. Each dimension of the FV representation is given by power at the wavenumber, $k_n$:

$$P(k_n) = \frac{1}{2\pi k_n} \oint_{C_n} |F(k_x, k_y)|^2 \ ds, \tag{A.14}$$

  where $C_n$ is the circle given by $\sqrt{k_x^2 + k_y^2} = k_n$ and $F(k_x, k_y)$ is the discrete Fourier transform of the two-dimensional field. To exclude the DC component, the term corresponding to $k_x = k_y = 0$ is explicitly omitted. We then converted $P(k_n)$ to log base 10.
- SINDyCP (SCP): SINDyCP (Nicolaou et al., 2023) is a sparse regression method for fitting differential equations to data. We used it in its "weak" formulation, whereby equations are fit to patches of data integrated against test functions. We used 500 subdomain patches and a sequentially-thresholded least squares (STLSQ) optimizer with threshold $1 \times 10^{-2}$.

In all cases, TRENDy used a four-layer MLP with rectified nonlinearities for its NODE module. This NODE was run with $dt = .01$ for total duration of $T = 1.0$. Note that since the number of time steps in TRENDy's solution and the true solution were not always the same, we co-registered the corresponding time series and omitted intervening time steps during loss computation. For the Brusselator experiment, we set the derivative regularizer to be $\beta = 10^{-4}$ and it was 0 otherwise. We used an Adam optimizer (Kingma & Ba, 2014) with learning rate $10^{-4}$.

All numerical continuation experiments were performed using pseudo-arclength continuation with a Newton-Raphson correction having threshold $1 \times 10^{-5}$. In parallel, a bisection algorithm with 15 steps was used to detect bifurcations using an eigenvalue threshold of $1.0$.

|  | No Noise | Boundaries | Patches |
|---|---|---|---|
| TRENDy Train | $4.23 \times 10^{-1}$ | $7.61 \times 10^{-1}$ | $9.44 \times 10^{-1}$ |
| TRENDy Test | $4.64 \times 10^{-1}$ | $7.59 \times 10^{-1}$ | $1.01 \times 10^{0}$ |

Table A.1: Mean square error between true and predicted reduced-order dynamics in the reduced domain (TRENDy).

| Holdout ($\epsilon$) \ Noise | No noise | Boundaries | Patches |
|---|---|---|---|
| $\epsilon = .015$ | $4.34 \times 10^{-4}$ | $7.94 \times 10^{-4}$ | $1.05 \times 10^{-3}$ |
| $\epsilon = 0.5$ | $8.91 \times 10^{-4}$ | $1.11 \times 10^{-3}$ | $7.85 \times 10^{-3}$ |
| $\epsilon = 1.0$ | $1.01 \times 10^{-3}$ | $4.25 \times 10^{-3}$ | $1.05 \times 10^{-2}$ |

Table A.2: TRENDy's Forecasting error across noise and holdout ($\epsilon$) conditions.

**Gray Scott.** Data were generated with parameters $F, k$ drawn uniformly on $[.045, .055] \times [0.0, .075]$. For $F = .05$, the true bifurcation occurs at $k^* = .062$. We generated 1000 training samples with $|k - k^*| > .01$ and 250 testing samples with $|k - k^*| < .01$.. Samples were generated by initializing $(u, v)$ uniformly at $(1, 0)$ plus pointwise gaussian noise of mean 0 and standard deviation .01. The initial condition was thresholded at 0. Solutions were produced from a forward Euler scheme with $dt = 1$, $dx = 1.5$ and which was solved for $T = 4500$ steps. The size of the spatial domain was $64 \times 64$.

For the bifurcation experiment, TRENDy was fit over 2000 epochs, each of which took approximately 31.48 seconds. We used a burn-in period of 10 time steps. RMSE results are in Table. A.1

For the classification experiment, we generated 5000 samples from within the testing regime and then performed a new 80/20 train/test split with class balancing. Ground truth labels were created by 4-way k-means clustering on the full scattering spectrum. We confirmed these were visually sensible groupings and that they matched standard analyses of the Gray Scott model (Munafo). TRENDy was then fit over 5000 epochs to each feature type (SG, T2-10, FV). SINDyCP learns an explicit equation for the PDE in the observable space, and, as we wished to evaluate reduced feature representations of the data, it was not appropriate as a benchmark. TRENDy's predicted steady state for each feature condition was used as a representation of the data to be then classified by a support vector machine (SVM). We used sklearn's model out-of-the-box with no modifications.

**Brusselator** We used 2000 training and 500 testing samples in the parameter regions described in the main text and controlled by the holdout factor, $\epsilon$. Solutions were produced with $dt = .01$ and were run until $T = 20$. RMSE on testing data for different holdout and noise values are found in Table A.2.

**Lizard patterning** Data was acquired by randomly sampling 100 locations in the high-resolution, 300-frame video of the developing lizard. We took the $d = 10$ most activate coefficients for training. Dynamics were fit in log scale in order to account for natural scale differences between zeroth, first and second-order coefficients.

Classification was performed on an 80/20 split of TRENDy's final state using an out-of-the-box sklearn SVM which had an $L^2$ regularizer of $C = 1.0$.

**The Ising model.** The Ising model is a simple, standard model of magnetism from statistical physics (Shekaari & Jafari, 2021). It has been applied widely in the physical and life sciences, notably in recent work on spatial patterning in living matter (Zakany et al., 2022). Unlike other systems investigated in the current study, the Ising model is stochastic and discrete-time. Nevertheless, TRENDy can still be used to fit the effective behavior of this model, which we demonstrate here.

A typical setting for the two-dimensional Ising model consists of a large grid of spins $S = s_i{}_{i=1}^n$ where each $s_i$ can assume a state in $\{-1, 1\}$. The energy associated to this configuration is

$$E(S) = J \sum_{\langle i,j \rangle} s_i s_j, \tag{A.15}$$

where $J$ is a scalar and the sum is taken over nearest neighbors. The probability of finding the system in a given configuration is

$$P(S) \propto e^{-\frac{E(S)}{k_B T}}, \tag{A.16}$$

where $k_B$ is Boltzmann's constant, $T$ is a temperature scalar, and the constant of proportionality is given by the so-called partition function.

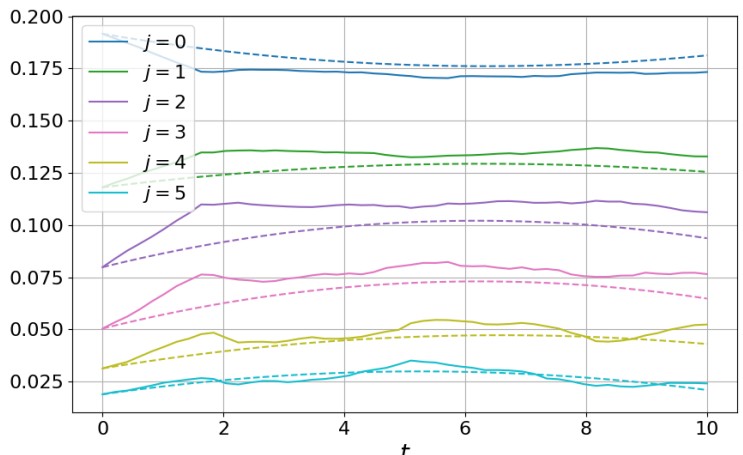

Figure A.2: *Example fit of Ising scales.* Solid lines depict the evolution of six scattering scales. TRENDy fit in dashed lines with corresponding colors. Time arbitrarily scaled to 10 total units. The temperature for this sample was $T = 6.46$.

There exists a critical temperature, $T_{crit}$ at which the system tends in the long term towards a magnetized state in which spins align on average. Above this temperature, magnetization is zero and correlations between spins decay exponentially as a function of their distance in the grid. However, as one approaches the critical temperature, these correlations tend to increase, resulting in the formation of so-called magnetic domains, i.e. regions of aligned spins. We should expect these domains to grow larger in scale the closer one gets to the critical temperature.

We sought to examine if TRENDy could learn the relation between temperature and predict domain size. We generated 1000 simulations of a $64 \times 64$ Ising model which was simulated for 100 timesteps using MCMC. Temperatures were chosen uniformly randomly on $[4.5, 7.5]$, a range above the critical temperature. We

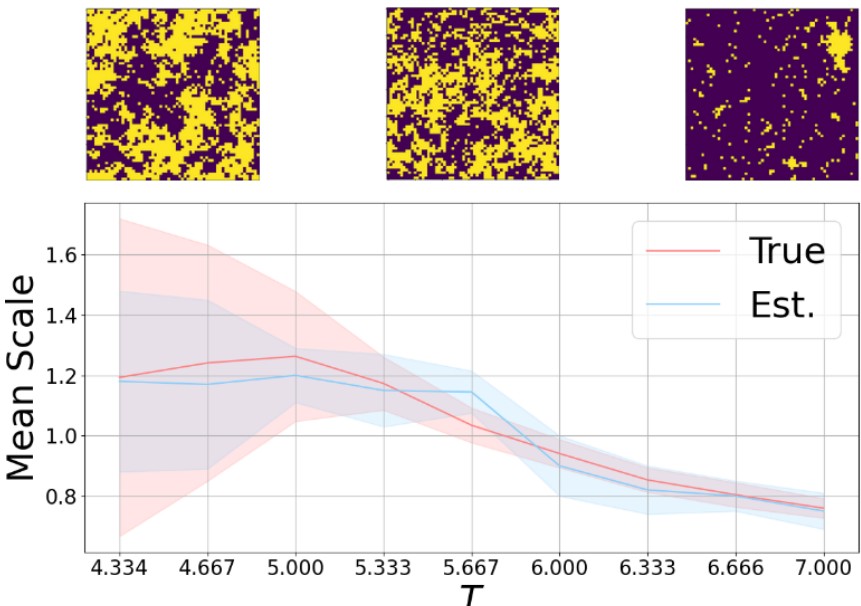

Figure A.3: *Scale versus temperature*. The average scale present after the evolution of the system was measured across a range of temperatures. Mean values in ten bins from $4.5$ to $7.5$ were measured directly from scattering scales of the Ising model and are plotted in red with standard deviation in the shaded region. The TRENDy estimate is in blue. Upper insets show samples Ising configurations at three temperature values from left to right: $4.5$, $6.0$ and $7.5$.

expect domain size to grow near the left end of this interval. As we expect domain size to be reflected in first order spatial statistics, we restricted the scattering measurements to first order coefficients. Further, as there is complete rotational symmetry in the Ising model, we further averaged over all orientations. That is, TRENDy's effective state was given by

$$a_j(t) = \frac{1}{L} \sum_{\ell=1}^{L} |u(\cdot, t) \star \psi_{\ell,j}| \star \phi \tag{A.17}$$

for $L = 8$ orientations and $j = 1, \ldots, 6$ scales, since $2^6 = 64$ was the size of the grid. Each of the 1000 Ising samples was thus converted into a 100 time-step, 6 dimensional time series representing the evolution of these scales. These time series were quite noisy, so we smoothed them with a moving average having window size 10.

TRENDy was fit to 800 of these samples with temperatures drawn uniformly from the aforementioned range (e.g. Fig. A.2). On the test data, we measured the average scale of TRENDy's prediction at the end of its evolution at time $T$. This average scale was given by $\frac{1}{c} \sum_j j a_j(T)$ for $c = \sum_j a_j(T)$. As expected, we find an inverse relation between temperature and average scale (Fig. A.3, red), indicating the formation of magnetic domains nearer the critical temperature. This trend is fit closely by TRENDy (blue). Notably, we see variance in average scale also increase at lower temperatures. This is to be expected since, near the critical temperature, correlation length diverges and no single scale dominates.

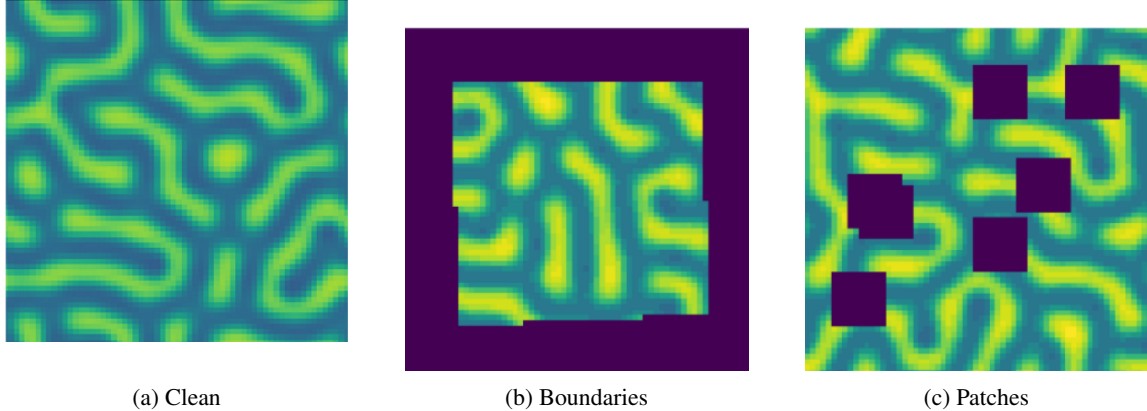

(a) Clean       (b) Boundaries       (c) Patches

Figure A.4: *Gray Scott samples with noise*.

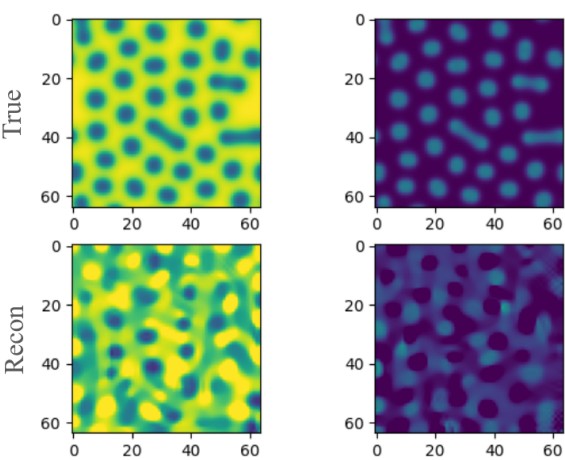

Figure A.5: *Example reconstruction: spots*. Reconstruction made from minimizing $\|S(u) - S(u_{est})\|_2$ where $u_{est}$ was initialized as white noise. Reconstruction was based on all scattering coefficients: $L = 8$ orientations and $j = 6$ scales up to order 2. We tried fitting these coefficients directly with TRENDy, but bifurcation prediction was substantially worse as a result. Only the T2, T5, T10 cases were both predictive of bifurcations and patterning class.

### A.1.6 COMPUTE DETAILS

TRENDy was trained with an Adam optimizer (Kingma & Ba, 2014) and with the Kymatio implementation of scattering (Andreux et al., 2020). Synthetic PDE data and the TRENDy NODE solutions were both acquired by forward Euler schemes (details for each system appear in the main text). TRENDy was trained in PyTorch (Paszke et al., 2019) and numerical continuation was run on BifurcationKit (Veltz, 2020) in Julia (Bezanson et al., 2017).

### A.2 SUPPLEMENTARY FIGURES

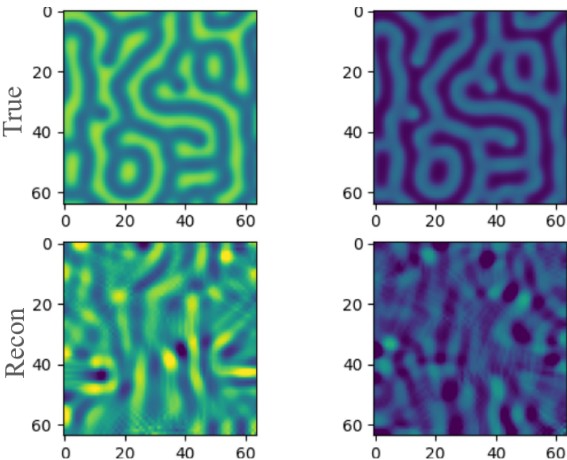

Figure A.6: *Example reconstruction: stripes*. Reconstruction made from minimizing $\|S(u) - S(u_{est})\|_2$ where $u_{est}$ was initialized as white noise. Reconstruction made from minimizing $\|S(u) - S(u_{est})\|$ where $u_{est}$ was initialized as white noise.

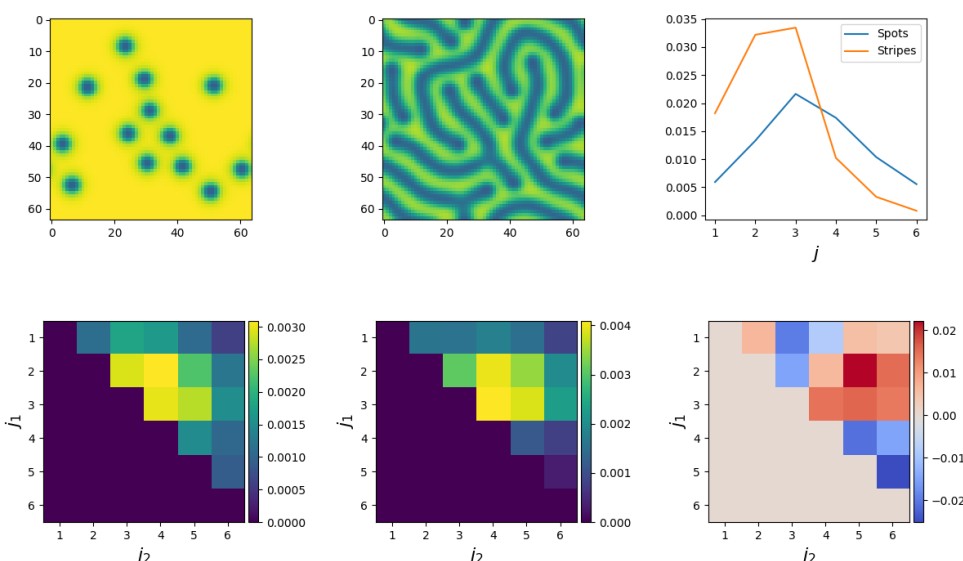

Figure A.7: *Scale comparison between two Gray Scott samples. (Top Row)*: Spots, and stripes images shaped $64 \times 64$, sampled from class 1 and 4 respectively (see main text). Scattering transform was performed with $L = 8$ orientations and $J = 6$ scales. Rightmost panel: average activity in each of the six scales in first order coefficients, where the average is taken across orientations. Both patterns have the same dominant spatial frequency (which is controlled by the diffusion constants, identical for these systems), but the dispersion of frequencies is different: spots have more power in higher scales; stripes, more power in lower. *(Bottom row)*: Second order scattering coefficients, again averaged over orientation. Arrays show activations in coefficients generated by filtering first at scale $j_1$ and then at scale $j_2$. First two panels correspond to the patterns above (spots/stripes). Rightmost panel is a difference between these activations, which indicates that the $j_1 = 2$, $j_2 = 5$, is a distinguishing feature. This makes sense since the spots spectrum is overall flatter (so that small scales and large scales have similar activations).

