# OpenReview forum: "TRENDy: Temporal Regression of Effective Nonlinear Dynamics"
_ICLR.cc/2025/Conference — ICLR 2025 Poster_

### Official Review · Reviewer_Lune · 2024-10-31

**Soundness:** 2
**Presentation:** 3
**Contribution:** 2
**Rating:** 6
**Confidence:** 5

**Summary:**

The paper introduces TRENDy as a data-driven, low-dimensional predictive modeling approach for spatiotemporal dynamics. TRENDy uses multiscale filtering to map data into effective dynamics, then fits a NODE to this reduced space. This process robustly regularizes the NODE phase space, making TRENDy resilient to noise. The so-called TRENDy is examined on synthetic and real data across physical and life sciences. It's posed as a versatile tool for studying and controlling spatiotemporal dynamics.

**Strengths:**

N/A

**Weaknesses:**

The significance of the work is missing by authors specifically compared to the state of the art approaches. In addition, the complexities of fitting by a neural ordinary differential equation need to be justified.

**Questions:**

The main question is comparing with the outstanding class of operator theoretic settings.

---

> ### Author Response · Authors · 2024-11-24
> **Response to R3**
>
> We would like to thank the author for these remarks. We have taken steps to clarify how our approach differs from existing methods both in premise and implementation. Crucially, unlike various state of the art approaches to solving forward problems in PDEs, including via neural operators (like the study by Li 2021 which we now cite in the revised manuscript), we are not specifically interested in fine-grained spatial prediction. Indeed, as we now show in several benchmark studies, learning a full spatial model, as with SINDyCP can lead to somewhat accurate bifurcation prediction (Fig. 2, rightmost data) but only at the cost of sensitivity to strong noise.
>
> One of the principal benefits of learning a reduced-order model is the robustness which we have quantitatively demonstrated (Fig. 2, right panel). Further, we now show in several decoding and interpretability experiments (Figs. 3, A.5, A.6, A.7) that these latent representations of the visible dynamics still suffice to decode meaningful aspects of the input data. In short, we see our approach as one end of a robustness-prediction tradeoff, and our experiments explore how this robustness can actually be used to decode high-level properties of the underlying dynamics better than state of the art forward solving models. Naturally, future work could explore this tradeoff in greater detail.
>
> We agree with the reviewer that details about training the neural ODE were not clear enough in the original manuscript. Training details are now included in full in the appendix (Sec. A.1.5).

---

### Official Review · Reviewer_MJPy · 2024-11-03

**Soundness:** 3
**Presentation:** 4
**Contribution:** 3
**Rating:** 6
**Confidence:** 3

**Summary:**

The framework introduced by the authors captures the dynamics of spatio-temporal parameterised PDE systems by relying on established reduced order modelling techniques and training a NODE solver. The author(s) showcase that the framework can capture the bifurcation points in various test and real datasets.

**Strengths:**

- The paper is well presented and clearly structured and introduces a framework which could prove to be impactful in the area of parameterised PDE modelling.
- The authors have included a strong baseline model (SINDyCP) for the Gray Scott example.

**Weaknesses:**

- The lack of decoder reduces the interpretability of the approach. This point is highlighted by the authors themselves but should be made earlier in the work and ways in which it will be tackled should be outlined.
  - All references should be checked, there are various instances of spelling mistakes.
- The addition of legends to Figure 3 would improve the presentation of the results.

**Questions:**

- Did the authors trial any alternatives to the scattering transforms? To showcase the touted benefits of the chosen approach.
- Is there a reason the baseline is not implemented for The Brusselator and real-world examples?
  - Have alternative types of bifurcations being considered (rate-induced bifurcations)?

---

> ### Author Response · Authors · 2024-11-24
> **Response to R2**
>
> We thank the reviewer for the constructive remarks. We have made numerous updates to the manuscript which address all of the reviewer’s comments. Notably, we now include several interpretability and decoding experiments which show that the TRENDy’s effective state still includes low-level information about the original state space without an explicit, learned decoder. We demonstrate that with only a few effective dimensions, we can decode pattern class at high accuracy (Fig. 3), an important problem in reaction-diffusion modeling. We also show that as the number of effective dimensions grows large, one can reconstruct the input pattern (Figs. 10-11). What’s more, we show how scattering coefficients can be used to explain spectral differences between these steady state patterns (Fig. 12). Further, we now systematically compare the scattering transform to other features, like average spatial gradients and spectral features, against which our method compares favorably.
>
> In its revised version, following the reviewer’s comments, we have structured the paper so that the benchmarking now includes several other approaches (averaged spatial gradients, Fourier features, TRENDy with fewer coefficients) and focuses mostly on the Gray Scott case. Having benchmarked our approach in this way, the subsequent Brusselator example is meant to demonstrate specifically to evaluate our method across a huge swath of parameters (Fig. 4). The next experiments on the lizard data (and the new Ising case which we have added to the revised manuscript (Sec. A.1.5, Ising model), Figs. A.2, A.3) are meant to demonstrate yet further qualitative aspects of our approach which do not lend themselves neatly to standard quantitative benchmarking.
>
> The reviewer’s request for different types of bifurcations is well met. In order to demonstrate both a categorically new type of system and additional interpretable aspects of our framework, we have included a new supplementary section on fitting an Ising model (see supplementary section A.1.5, Ising model). This new experiment shows how our approach can be deployed without further adjustments to systems with very different physics compared to those explored earlier.
>
> Finally, we thank the reviewer for noting stylistic and presentation errors, which we’ve since corrected.

---

### Official Review · Reviewer_ahWW · 2024-11-04

**Soundness:** 2
**Presentation:** 2
**Contribution:** 2
**Rating:** 3
**Confidence:** 4

**Summary:**

The paper introduces TRENDy, a framework designed for learning low-dimensional surrogates of complex dynamical systems where underlying PDEs are unknown, and data is noisy or limited. The main contributions of TRENDy include:

1. Modeling Effective Dynamics: TRENDy maps underlying PDE using multiscale filtering into a reduced space and models the reduced representation with a neural ODE. This NODE captures the system's behavior based on its governing parameters, which enables TRENDy to predict system dynamics in new parameter spaces.

2. Predicting Bifurcation: moreover, for a parameter-dependent system, the framework is able to predict bifurcations, where a sudden qualitative change in its behaviors. TRENDy also shows robustness to noise in bifurcation localization.

3. Application to Real-World Data: authors also used the patterning in the ocellated lizard as an example to illustrate how the framework's latent space captures meaningful biological features.

**Strengths:**

1. Originality:

The approach that combines scattering transform and neural ODEs to model the effective dynamics is novel, especially given its application to bifurcation prediction, a challenging task where data is limited and governing equations are unknown.

2. Quality and Clarity:

The paper shows rigorous methodology and fruitful details in various experiments. Explanations on filtering operations, the NODE structure, and training details led the model's design to be crafty and reliable.

3. Significance:

TRENDy addresses a crucial question in modeling systems governed by unknown or complex PDEs, where direct analytic solutions are impractical to get. The framework’s adaptability to new parameter spaces may also have numerous applications in real-time system control and scenario exploration. In a nutshell, the authors have shown that TRENDy has the potential to significantly advance research in fields like synthetic biology, physics, climate change and ecology, where such questions regarding complex dynamical systems are pretty common.

**Weaknesses:**

1. Multiscale Filtering:

The use of multiscale filtering (e.g., scattering transforms) is central to TRENDy, while the specific choice and design of the filtering process are not fully explored in the paper. Authors should provide more why they prefer this type of dimension reduction technique rather than others (for example, do ablation studies on other type of techs and show the one you mentioned is the best). Moreover, compared with too  many experimental details in the main text (better go to supporting materials), it is necessary to say more on multiscale filtering details, e.g. effects of the choice on scattering coefficients. Such explanations / experiments are essential to keep novelty of the paper, since they are numerous papers working on PDE + DL topics (and some of them should be acknowledged, e.g. PDE-net by Long et al. [1], PINNs by Karniadakis et al. [2], and other papers focusing on effective dynamics, see [3] and [4]).

2. Reconstructing State Space:

Just like lifting and restriction in the equation-free approach, TRENDy should have the module which maps the latent dimensions back to the full PDE state space. Without such an explicit decoder, the ability to verify the reduced dynamics against full state predictions will be limited. It will also become an obstacle for researchers in other fields to explore the explainability by utilizing your model. It seems adding a mechanism for decoding reduced dynamics back into full spatial states or maybe explaining why this is not feasible in your scope is essential.

3. Miscellaneous:

I suggested the reviewers consider the following issues, and if time allows, do some elaboration.

a) Extending the experimental scope (e.g. systems with chaotic attractors, or discrete-time systems).

b) Discussing the model’s performance on large datasets and its computational demands in both training and inference.

c) Implementing interpretability techniques (e.g., parameter sensitivity, feature importance) to provide insights on multiscale filtering.

[1] https://arxiv.org/pdf/1710.09668

[2] https://arxiv.org/pdf/1711.10561

[3] https://www.nature.com/articles/s41467-024-48024-7

[4] https://pubs.aip.org/aip/cha/article-abstract/34/6/063128/3298062/Tipping-points-of-evolving-epidemiological?redirectedFrom=fulltext

**Questions:**

Besides several concerns that mentioned in the weakness part, here are several questions regarding the paper details:

Figure 1: what is $S_i(0)$ here? And why do they have different heights?

Eq 1: it is better to use $u(x, y)$ rather than $u(r)$, as you are talking about 2D space now.

Usage of subscripts (Line 138 and other notations): the subscripts sometimes are very misleading. e.g. $u_{\theta}$ and $u_0$.

Line 139: for $u_0 \notin D$, do you mean interpolation and/or exploration?

Line 141: needs explanation of what $U$ is.

Line 148: similar as what mentioned previously, why do you assume $\Phi$ is hardwired and unlearned? Can the multiscale filtering parameters be learnable?

Figure 2: this figure needs more details to explain. For example, you should say the inset squares means PDE solutions (otherwise it is misleading).

Line 207: the approximately equal symbol here is incorrect. And moreover, what is SINDyCP? Formula of it? Does it have any assumptions? Have you cited it?

Lines 230-232: may need a figure to illustrate your conditions. For example, what is “patches”?

Line 266: need to show why $S_{1, 2}$ almost equals to $<u>$ and $<v>$.

Line 306: use def eq symbol “:=” here in $d_{\gamma} (\theta)$.

---

> ### Author Response · Authors · 2024-11-24
> **Response to R1**
>
> We thank the reviewer for acknowledging the strengths of our approach and for the numerous constructive remarks. The reviewer has rightly asked us to justify our lack of decoder as well as our use of scattering for dimensionality reduction. As we now demonstrate in the manuscript, scattering actually circumvents the need for an explicit decoder. Indeed, our premise is that often the effective dynamics are  the dynamics of interest, while fine spatial details are nuisances to be abstracted away. Consider the lizard example: the goal of systems biologists studying patterning is generally not to predict the color of each individual scale, but rather to predict the general class of patterns an animal might produce (and how this process occurs).
>
> A good effective model should (1) predict if there are different effective classes and (2) ensure that these classes are interpretable in terms of physical structures in the full state space.  Our choice of the scattering transform was motivated by these two point. As we now show in several experiments, scattering coefficients can not only be used to predict bifurcations, but also can also be used to explain the difference between the spatial patterns (Fig. 3, Fig. A.7) that emerge during these bifurcations. And, in the limit of many coefficients, we also demonstrate how these coefficients can be decoded into actual images (Fig. A.6, A.6).
>
> Crucially, this focus on effective dynamics distinguishes our work fundamentally from that of Long, Karniadakis and others. As these studies seek to accelerate the solutions of traditional forward or inverse PDE problems with data, they naturally make use of a reconstruction loss, which is often regularized with prior knowledge of physical laws. Our commitment to effective dynamics allows us to sidestep full reconstruction and thereby use no prior knowledge of underlying physics, which is often unavailable.Notably, none of these frameworks is parametric: they fit a solution to one equation and do not consider the bifurcation problem. TRENDy is designed to this end and the universal expressiveness of the scattering transform means that spatial structures in these many different parametric settings can be compactly represented with no retraining. Other frameworks which are parametric (SINDyCP) are in turn not effective. Our approach, in short, is a synthesis of the parametric and effective methods.
>
> To emphasize these points and to answer the reviewer’s remarks regarding decoding, interpretability and the use of the scattering transform, we have added the following new experiments:
>
> 1. A classification experiment (Sec. 4, Fig. 3) whereby TRENDy’s learned state can be used to decode patterning classes of the Gray Scott model. Again, as our focus is not to reconstruct exact states but rather more qualitative patterning behaviors, it suffices to show that we can decode visually meaningful pattern classes from TRENDy’s dynamics on test data. This demonstrates:
>
> 1a. TRENDy can indeed map back to the observable space in a categorical, instead of pixelwise, sense (Fig. 3).
>
> 1b. Scattering and its hardwired features helps explain these classes in terms of specific structures (Fig. A.7). This is demonstrated with a straightforward analysis of which spatial scales and interactions between scales are predicted as a function of Gray Scott parameters.
>
> 1c. Increasing the number of scattering parameters helps facilitate an exact reconstruction of the input pattern at the cost of worsened bifurcation prediction (Fig. A.6, A.7). This tradeoff highlights effective dynamics as a theoretically different aim from exact spatial prediction.
>
> 2. A comparison between the scattering transform and new interpretable baseslines: average spatial gradients, Fourier features, and TRENDy with different numbers of dimensions (Figs. 2, 3).
>
> 3. A new experiment where TRENDy is fit to an Ising model with a varying temperature (Sec. A.1.5, Ising Model). This addresses both the reviewer’s request for new systems (Ising is stochastic and discrete-time) and the request for interpretability studies. We provide a systematic analysis of the dominant spatial scales predicted by TRENDy as temperature decreases towards the critical point (Fig. A.3).
>
> We have also made numerous stylistic and graphical adjustments following the reviewer’s detailed remarks, notably:
>
> 1. New citations and explanations of related methods (Intro).
> 2. Revamped Fig. 1.
> 3 Notation has been cleaned overall, including all suggestions made by the reviewer.
> 4. Clarification that focus on the (still challenging) interpolation problem.
> 5. Fig. 2 is now simplified and includes benchmarks.
> 6. Appropriate citation and description of SINDyCP.
> 7. Experimental detail has now been moved to the appendix, including average compute times for training epochs, sizes of train vs test data, etc.
> 8. Examples of noisy data are given in Fig. A.4
> 9. We note potential learning of features in the discussion section.

---

### Author Response · Authors · 2024-11-23
**Thanks and overall remarks**

We thank all reviewers for their constructive remarks, which we believe greatly strengthen the manuscript. Our study introduces TRENDy, a method for learning effective, reduced-order dynamics from spatiotemporal data. The original manuscript showed how our combination of multiscale filtering, NODE dynamics and parametric modeling was a powerful model of bifurcation detection, a problem of general relevance across the sciences. The reviewers have recognized the importance of this problem setting and the value of our approach to solving it. In light of the reviewers’ comments, we have made numerous additions to the manuscript which clarify–through both changes in language and additional experiments–all components of the TRENDy pipeline. In the resubmitted manuscript, key changes are highlighted in blue.

Two important concerns of reviewers were (1) our choice to not use an explicit decoder and (2) our decision to use the scattering transform for representing spatiotemporal dynamics. As we argue in the following comments and in the revised manuscript, our focus on effective dynamics actually circumvents the need for an explicit decoder, as long as the effective state is still informative about the original state space. Indeed, the scattering transform gives us this informativeness, as we now show in several new decoding and interpretability experiments (Figs. 3, A.3. A.5, A.6, A.7). This is further justified with new benchmark experiments which demonstrate the scattering transform’s performance over and above competing feature representations.

In addition to numerous stylistic and organizational changes, we have made the following significant adjustments:

1. We have shown how TRENDy’s effective state can be decoded into categorical (Fig. 3) and fine-grained spatial information (Figs. A.5,6,7) about the input PDE without need for an explicit learned decoder.

2. We have benchmarked (Fig. 2, right) and otherwise justified our use of the scattering transform, including by showing how TRENDy’s state can explain the emergence of spatial structures at particular scales (Fig. A.7)

3. We have shown that TRENDy can be applied with no further adjustments to the Ising model, which is both stochastic and discrete-time, a contrast with our further experiments. We have also demonstrated how TRENDy’s effective state on Ising dynamics can be interpreted in terms of the growth of magnetic domains at specific spatial scales.

---

### Author Response · Authors · 2024-12-01
**Request for further discussion**

We thank the reviewers once again for their helpful remarks. In our recent replies, we endeavored to address each point raised by the reviewers. The revised manuscript features several wholly new experiments which demonstrate the performance and interpretability of the proposed method. We would welcome continued discussion with the reviewers and will be happy to address any remaining points before the rebuttal period ends.

---

### Meta-Review · Area_Chair_Bx4X · 2024-12-23

**Metareview:**

This paper is on modeling spatially extended PDEs by first mapping input data to a low-dimensional space using scattering transforms, and then employing Neural ODEs to to fit the resulting "effective dynamics".  This method can predict system behaviors such as bifurcations, which is validated on a few examples of spatio-temporal dynamics.

Strengths: Coherent formulation for modeling effective Dynamics; shows robustness to noise in bifurcation localization; application to Real-World Data.

Weaknesses: Primarily two concerns were raised: 1) lack of interpretibility and not learning an explicit decoder and 2) motivating the use of scattering transform as opposed to an alternative for representing spatiotemporal dynamics.

**Additional Comments On Reviewer Discussion:**

The authors added new decoding and interpretability experiments & comparison of the scattering transform’s performance against competing feature representations.  It is also shown that effective low-dimensional state still includes low-level information about the original state space without an explicit, learned decoder.  The use of Neural ODEs with Scattering transforms may be of interest to the PDE modeling community. Reviewer ahWW raised the most significant concerns, but also acknowledged the papers novelty particularly on bifurcation prediction;  that model design is "crafty and reliable" and that the method "has the potential to significantly advance research in fields like synthetic biology, physics, climate change and ecology, where such questions regarding complex dynamical systems are pretty common". Overall, the reviews lean towards acceptance which weighs into the final decision.

---

### Decision · Program_Chairs · 2025-01-22

Accept (Poster)